# Network-based multi-omics integration reveals metabolic at-risk profile within treated HIV-infection

**Flora Mikaeloff[1]\*, Marco Gelpi[2], Rui Benfeitas[3], Andreas D Knudsen[2], Beate Vestad[4,5], Julie Høgh[2], Johannes R Hov[4,5,6], Thomas Benfield[7], Daniel Murray[8], Christian G Giske[9], Adil Mardinoglu[10,11], Marius Trøseid[4,12], Susanne D Nielsen[2]†, Ujjwal Neogi[1]\*†**

[1]The Systems Virology Lab, Division of Clinical Microbiology, Department of Laboratory Medicine, Karolinska Institute, Stockholm, Sweden; [2]Copenhagen University Hospital Rigshospitalet, Copenhagen, Denmark; [3]National Bioinformatics Infrastructure Sweden (NBIS), Science for Life Laboratory, Department of Biochemistry and Biophysics, Stockholm University, Stockholm, Sweden; [4]Research Institute of Internal Medicine, Oslo University Hospital Rikshospitalet, Oslo, Norway; [5]Norwegian PSC Research Center, Oslo University Hospital Rikshospitalet, Oslo, Norway; [6]Institute of Clinical Medicine, University of Oslo, Oslo, Norway; [7]Department of Infectious Diseases, Copenhagen University Hospital – Amager and Hvidovre, Hvidovre, Denmark; [8]Centre of Excellence for Health, Immunity and Infections (CHIP), Rigshospitalet, University of Copenhagen, Copenhagen, Denmark; [9]Division of Clinical Microbiology, Department of Laboratory Medicine, Karolinska Institutet, Stockholm, Sweden; [10]Science for Life Laboratory, KTH - Royal Institute of Technology, Stockholm, Sweden; [11]Centre for Host-Microbiome Interactions, Faculty of Dentistry, Oral & Craniofacial Sciences, King's College London, London, United Kingdom; [12]Institute of Clinical Medicine, Oslo, Norway

**\*For correspondence:**
flora.mikaeloff@ki.se (FM);
ujjwal.neogi@ki.se (UN)

†These authors contributed equally to this work

**Abstract:** Multiomics technologies improve the biological understanding of health status in people living with HIV on antiretroviral therapy (PWH). Still, a systematic and in-depth characterization of metabolic risk profile during successful long-term treatment is lacking. Here, we used multi-omics (plasma lipidomic, metabolomic, and fecal 16 S microbiome) data-driven stratification and characterization to identify the metabolic at-risk profile within PWH. Through network analysis and similarity network fusion (SNF), we identified three groups of PWH (SNF-1–3): healthy (HC)-like (SNF-1), mild at-risk (SNF-3), and severe at-risk (SNF-2). The PWH in the SNF-2 (45%) had a severe at-risk metabolic profile with increased visceral adipose tissue, BMI, higher incidence of metabolic syndrome (MetS), and increased di- and triglycerides despite having higher CD4+ T-cell counts than the other two clusters. However, the HC-like and the severe at-risk group had a similar metabolic profile differing from HIV-negative controls (HNC), with dysregulation of amino acid metabolism. At the microbiome profile, the HC-like group had a lower α-diversity, a lower proportion of men having sex with men (MSM) and was enriched in Bacteroides. In contrast, in at-risk groups, there was an increase in *Prevotella*, with a high proportion of MSM, which could potentially lead to higher systemic inflammation and increased cardiometabolic risk profile. The multi-omics integrative analysis also revealed a complex microbial interplay of the microbiome-associated metabolites in PWH. Those severely at-risk clusters may benefit from personalized medicine and lifestyle intervention to improve their dysregulated metabolic traits, aiming to achieve healthier aging.

## Editor's evaluation

This important study systematically integrates convincing multi-omics data to identify the metabolic at-risk profiles within people living with HIV on antiretroviral therapy and presents findings that have focused importance and scope. The authors have used appropriate and validated methodology in line with the current state-of-the-art and have produced a paper that is of great interest to a specialised audience interested in HIV infection and metabolic mechanisms.

## Introduction

Antiretroviral therapy (ART) has improved the immune profile by suppressing viral replication and reducing the morbidity and mortality of people living with HIV (PWH). Yet living with HIV under ART induces a strong metabolic perturbation in the body due to virus persistence, immune activation, chronic low-grade inflammation, and treatment toxicity, mostly with older antiretrovirals (*Yoshimura, 2017*). The biological shifts due to a mixed effect of drugs and viruses are also highly personalized depending on the patient genetic background, age, sex, immunological, and lifestyle factors (*Pelchen-Matthews et al., 2018*). Long-term HIV infection, even with viral suppression, is associated with an accentuated onset of non-AIDS-related comorbidities (*Deeks, 2011*). Consequently, diseases of the aged population appear in relatively young HIV patients, including cardiovascular disease, liver-kidney disease, and neurocognitive and metabolic disorders (*Nasi et al., 2017*).

Systems biological analyses are valuable methodologies for systematically understanding pathology and identifying potential novel treatment strategies (*Karahalil, 2016*). Microbiome studies have provided enormous knowledge about the microbial association with HIV status, sexual practice, and gender (*Zhou et al., 2020*; *Gelpi et al., 2020*; *Noguera-Julian et al., 2016*), and the possible interplay between HIV-related gut microbiota, immune dysfunction, and comorbidities like metabolic syndrome (MetS), and visceral adipose tissue (VAT) accumulation (*Gelpi et al., 2020*). Our extensive metabolomics studies from three different cohorts from India (*Babu et al., 2019*), Cameroon (*Mikaeloff et al., 2022*), and Denmark (*Gelpi et al., 2021*) with more than 500 PWH have indicated disrupted amino acid (AA) metabolism in PWH with ART (PWH) following prolonged treatment that plays the central role in the comorbidities such as MetS (*Gelpi et al., 2021*).

The application of integrative omics to understand the disease pathogenesis in PWH under suppressive ART is lacking. To our knowledge, no integrative omics studies have been performed to understand complex biological phenotypes in PWH during prolonged suppressive ART. Multi-omic characterizations may offer insights into understanding the mechanisms underlying biological processes in a specific disease condition. A recent longitudinal study integrating metabolomics, plasma protein biomarkers, and transcriptomics in patients' samples identified potential lipid and amino acid metabolism perturbations in PWH with immune reconstitution inflammatory syndrome (IRIS) (*Pei et al., 2021*). Our recent network-based integrative plasma lipidomics, metabolic biomarker, and clinical data indicated a coordinated role of clinical parameters like accumulation of visceral adipose tissue (VAT) and exposure to earlier generations of antiretrovirals with glycerolipids and glutamate metabolism in the pathogenesis of PWH with MetS (*Olund Villumsen et al., 2021*).

The present study aimed to identify a molecular data-driven phenotypic patient stratification using network-based integration of plasma metabolomics/lipidomics and fecal microbiota within a cohort of PWH with prolonged suppressive therapy who were at-risk of metabolic complications. We further investigated the underlying factors differing from these profiles and the link to their clinical phenotype to clarify the risk factors for metabolic disease.

## Results

### Comprehensive multi-omics characterization of PWH on successful cART

In this study, we used untargeted plasma metabolomics (877 metabolites) (*Gelpi et al., 2021*), lipidomics (977 lipids) (*Olund Villumsen et al., 2021*), and fecal 16 S rRNA microbiome [241 amplicon sequence variants (ASVs)] data (*Gelpi et al., 2020*) from 97 PWH from the Copenhagen Comorbidity (COCOMO) cohort (*Gelpi et al., 2018*) where we have three types of omics data available. Additionally,

**Table 1.** Patient characteristics.

| | Complete Cohort | SNF-1 | SNF- 2 | SNF-3 | P values |
|---|---|---|---|---|---|
| At-risk Classification | | HC-like | Severe at risk | Mild | |
| N | 97 | 19 | 44 | 34 | |
| Age in years, Median (IQR) | 54 (48–63) | 60 (48–68) | 54 (48–62) | 54 (51–60) | 0.75 |
| Gender, Male, N (%) | 84 (87) | 15 (79) | 40 (91) | 29 (85) | 0.36 |
| Ethnicity Caucasian, N (%) | 79 (81) | 15 (79) | 38 (87) | 26 (77) | 0.49 |
| Mode of transmission, N (%)<br>Homosexual/bisexual<br>Heterosexual<br>Other/unknown | 63 (65)<br>26 (27)<br>8 (8) | 9 (47)<br>7 (37)<br>3 (16) | 36 (81)<br>6 (14)<br>2 (5) | 18 (53)<br>13 (38)<br>3 (9) | 0.017 |
| CD4 Nadir, cells/mL, Median (IQR) | 235 (123–320) | 240 (127–330) | 240 (145–365) | 223 (42–290) | 0.49 |
| CD4 at ART Initiation, cells/mL, Median (IQR) | 287 (155–410) | 270 (120–360) | 318 (192–463) | 240 (108–320) | 0.11 |
| Viral Load at ART initiation, log copies/mL, Median (IQR) | 5.02 (4.34–5.61) | 4.87 (4.32–5.5) | 5.11 (4.74–5.61) | 4.94 (4.2–5.55) | 0.35 |
| CD4 at sampling, cells/mL, Median(IQR) | 713 (570–900) | 680 (540–958) | 762 (689–923) | 610 (475–819) | 0.015 |
| CD8 at sampling, cells/mL, Median (IQR) | 775 (600–1100) | 780 (630–879) | 894 (638–1300) | 700 (530–870) | 0.054 |
| Viral load (<50 copies/mL), N (%) | 97 (100) | 19 (100) | 44 (100) | 34 (100) | 1 |
| Duration of treatment in years, median (IQR) | 15 (9–18) | 15 (13–18) | 15 (8–18) | 14 (7–17) | 0.73 |
| Current Treatment, 1st drug, N (%)<br>ABC<br>TDF/TAF<br>Other | 31 (32)<br>42 (43)<br>24 (25) | 8 (42)<br>8 (42)<br>3 (16) | 13 (30)<br>19 (43)<br>12 (27) | 10 (29)<br>15 (44)<br>9 (27) | 0.84 |
| Current Treatment, 3rd drug, N (%)<br>NNRTI<br>PI/r<br>INSTI<br>Other | 38 (39)<br>18 (19)<br>15 (15)<br>26 (27) | 8 (42)<br>4 (21)<br>4 (21)<br>3 (16) | 14 (32)<br>11 (25)<br>6 (14)<br>13 (29) | 16 (47)<br>3 (9)<br>5 (15)<br>10 (29) | 0.45 |
| BMI, Mean (SD) | 24 (22–27) | 22 (19–25) | 26 (23–28) | 24 (22–27) | 0.003 |
| VAT, Median (IQR) | 89 (36–142) | 41 (19–106) | 127 (79–196) | 69 (26–100) | 0.0001 |
| SAT, Median (IQR) | 111 (70–167) | 69 (33–115) | 117 (82–174) | 119 (83–190) | 0.02 |
| MetS, N (%) | 43 (44) | 6 (32) | 31 (70) | 6 (17) | 0.000009 |
| Central obesity, N(%) | 57 (59) | 8 (42) | 32 (73) | 17 (50) | 0.033 |
| Waist circumference (cm) | 94 (87–101) | 90 (84–95) | 100 (91–105) | 90 (87–97) | 0.0007 |
| Hypertension, N (%) | 49 (51) | 5 (26) | 23 (52) | 21 (62) | 0.04 |

The online version of this article includes the following source data for table 1:

**Source data 1.** Comparative characteristics of the HC and PWH.

**Source data 2.** Non-significant characteristics between the cohorts.

we included 42 clinical and demographic features comprising lifestyle habits (food, medicine, alcohol, smoking), comorbidities linked to obesity and non-communicable chronic comorbidities (e.g. liver function, kidney function, and diabetes), and HIV-related measurements (viral load, treatment history, CD4 T-cell count, CD8 T-cell counts) (Appendix 1). The PWH were mainly male (86%, 84/97), of Caucasian ethnic origin (81%, 79/97), with a median (IQR) age of 54 (48-63) years. The median (IQR) duration of the treatment was 15 (9-18) years. At the time of sample collection, the viral load was below the detection level with successful immune reconstitution [median (IQR) CD4 T-cell count 713 (570-900) cells/µL] (*Table 1*). Additionally, 20 HIV-negative controls (HC) from the Danish population with similar sex proportions (90% male, 18/20) and median age (IQR) of 56 (50-67) years with slightly higher median (IQR) BMI 26 (23-29) compared to the complete cohort [24 (22-27), p=0.04; *Table 1—source data 1*]. The HC was used to reference multi-omics and define the HC-like PWH.

## Integrative omics-based similarity network fusion (SNF) identifies three clusters in PWH

To stratify the PWH based on their molecular signature, we used Similarity Network Fusion (SNF) that constructs similarity matrices and networks of PWH for each of the omics and fuses them into one

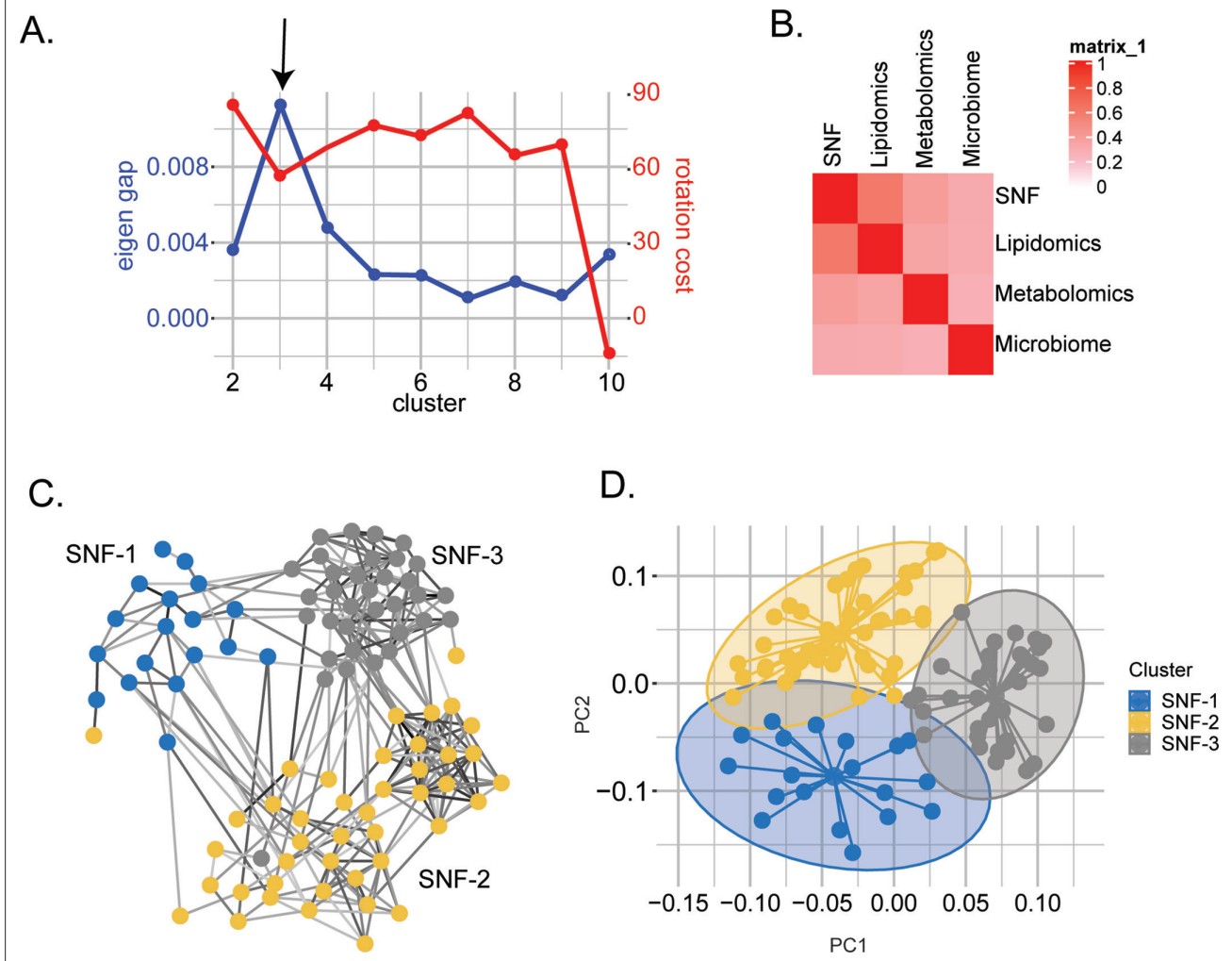

**Figure 1.** Similarity network fusion-based PWH stratification using lipidomics, metabolomics, and microbiome integration. (**A**) Scatter plot showing the maximization of Eigen gap and the minimization of rotation cost for optimizing the number of clusters. (**B**) Concordance matrix between the combined network (SNF) and each omics network based on NMI calculation (0=no mutual information, 1=perfect correlation). (**C**) SNF-combined similarity network colored by clusters (SNF-1/HC-like=blue, SNF-2/severe at-risk=yellow, SNF-3/mild at-risk=grey) obtained after spectral clustering. Edges' color indicates the strength of the similarity (black = strong, grey = weak). (**D**) PCA plot of samples based on fused network. Samples are colored by condition.

The online version of this article includes the following figure supplement(s) for figure 1:

**Figure supplement 1.** PCA plot of samples after prior standardization based on (a) Lipidomics (b) Metabolomics (c) Microbiome.

network that represents the full spectrum of the underlying data and disease status in PWH (*Wang et al., 2014*). We identified three clusters of patients, defined as SNF-1 (N=19), SNF-2 (N=44), and SNF-3 (N=34) (*Figure 1A*). The concordance matrix based on Normalized Mutual Information (NMI) score (0=no mutual information, 1=perfect correlation) showed that lipids had the most influence in the final network (NMI = 0.6), followed by metabolites (NMI = 0.4) and finally, microbiome (NMI = 0.3) (*Figure 1B*). Clear segregation of the SNF clusters (*Figure 1C*) was observed on the PCA plot based on the fused network values (*Figure 1D*) and PCA of single omics for lipidomics (*Figure 1— figure supplement 1A*) and metabolomics (*Figure 1—figure supplement 1B*) but not microbiome (*Figure 1—figure supplement 1C*).

## Cluster-specific clinical characteristics define a metabolic at-risk group

Cluster-specific clinical characteristics of PWH are presented in *Table 1*. Clusters were not statistically different for age, gender, duration of ART, and type of ART(p>0.05). On the other hand, SNF-1 had the healthiest profile (herein HC-like group), SNF-3 an intermediate (herein mild at-risk group, and SNF-2

the most severe metabolic perturbations herein severe at-risk group), indicating an at-risk metabolic profile. The severe at-risk group represented patients with high BMI, central obesity, higher VAT, and incidence of MetS (all p<0.05) but there was no association with measures of liver damage (alanine aminotransferase, ALT) or reduced kidney function (estimated glomerular filtration rate, eGFR), all p>0.05 (*Table 1—source data 2*). Regardless of disease severity, the severe at-risk group's patients had a higher CD4+ T-cell count at the time of sample collection and more men who have sex with men (MSM) as transmission mode compared to the other clusters (all p<0.05) considered as confounding factors here. The at-risk groups, severe and mild, had a significantly higher subcutaneous adipose tissue (SAT) and incidence of hypertension compared to the HC-like cluster (all p<0.05). The HC-like cluster had the lowest BMI, SAT, VAT, and incidence of hypertension (all p<0.05).

## Lipids and metabolites highlight clinical differences between patient clusters

Next, we performed the differential metabolite and lipid class abundance between the clusters. A similar lipid profile was observed between the HC-like, mild at-risk groups and HC (*Figure 2A and B*, and *Supplementary file 1*). Patients from the severe at-risk group showed a significant increase in diglycerides (DAG; *Figure 2A*) and triglycerides (TAG) (*Figure 2B*) compared to HC-like, mild at-risk cluster, and HC (all FDR <0.1) as well as other lipids classes which coordinate with their clinical metabolic profile (*Figure 2—figure supplement 1*). After adjusting for two confounders' modes of HIV transmission and CD4 count at sampling that are different between the clusters, the trends for lipid class remained the same (*Figure 2—source data 1*). In this analysis, the relation between cluster and ART class was not significant ($\chi^2$, FDR = 0.45). Still, we can mention that the three groups had an important proportion of missing data for this variable (16%, 29%, and 29%, respectively).

To identify the global metabolite impact on the cluster, we performed differential metabolite abundance (DMA) analysis. We kept stringent statistical parameters (FDR <0.005) and identified 159 metabolites with highly different metabolites among the groups (*Supplementary file 2*). The mild at-risk group and HC had only nine metabolites differing, in line with the high clustering of both groups shown with PCA (*Figure 2C*). The most perturbations were observed between HC and the HC-like PWH (124/159) and HC and severe at-risk group (62/159) (*Figure 2D*). Compared to HC, these clusters showed an up-regulation of the metabolites in the xenobiotics, nucleotides, and amino acid metabolism. In turn, the HC-like and severe at-risk groups showed similar metabolic profiles. Among these 159 metabolites, 50 had a low or moderate association with age and BMI (Spearman correlation, absolute $R$<0.4, p<0.1) and 51 with gender ($\chi^2$, p<0.1), showing the modest influence of individual characteristics on metabolomics profile. Within the PWH groups, after adjusting for the two confounders, the supervised principal component analysis of the significantly different metabolites (n=217) identified distinct clusters of HC-like, mild, and severe at-risk groups (*Figure 2E* and *Supplementary file 3*). The DMA identified the similarity of HC-like and severe at-risk groups with only 15 metabolites significantly different (FDR <0.05); most were part of lipid metabolism. Combining the in-depth metabolomics and lipidomic data indicated more personalized risk factors for PWH that the clinical features cannot explain. A complex interplay between the multi-omics layers defines overall health status.

## Sexual preferences influence the clusters' differences driven by the microbiome

As the metabolic aberrations were closely linked with the microbiome profile, we investigated the microbiome's impact on PWH clusters. The α-diversity indices indicated a loss of diversity according to Observed, ACE, se.ACE, Chao1, and Fisher indices in HC-like compared to the severe at-risk group (Mann Whitney, FDR <0.05; *Figure 3A*, *Figure 3—figure supplement 1* and *Figure 3—source data 1a*). A non-metric multidimensional scaling (NMDS) ordination of the dissimilarity-based index (Bray-Curtis) of diversity at the ASV level was performed to measure the inter-individual differences between groups (β-diversity; *Figure 3B*). Based on NMDS plot axis coordinate 1, the HC-like group was segregated separately from mild and severe at-risk groups (Mann Whitney, FDR <0.05, *Figure 3C*). The relative abundance of fecal microbiota was more influenced by the transmission mode than the cluster itself (*Figure 3—figure supplement 2A*). No other comorbidities on the microbiome profile were observed (*Figure 3—figure supplement 2B–D*). The severely at-risk group had a significantly higher

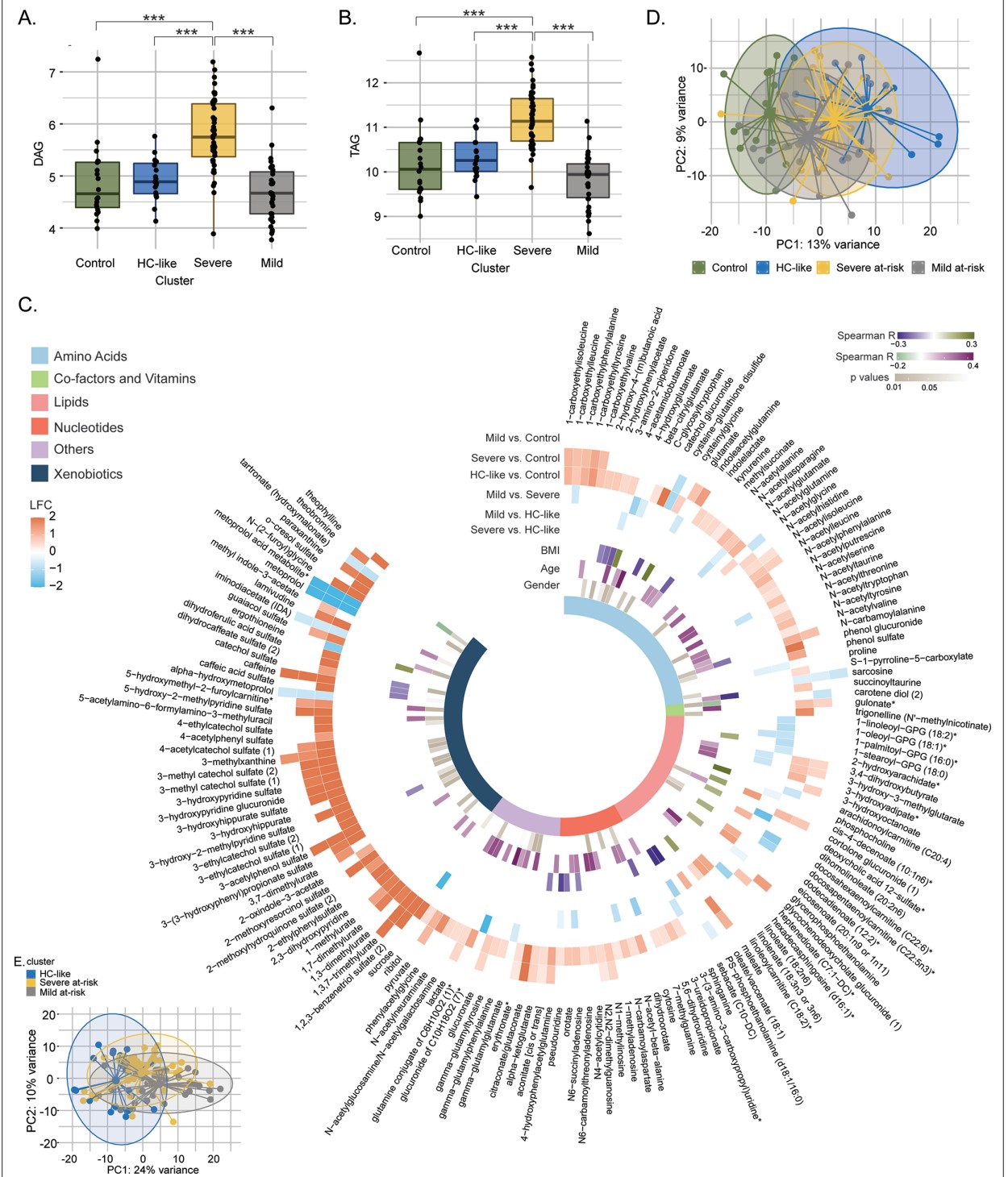

**Figure 2.** Lipidomics and metabolomics, characterization of the PWH clusters. (**A**) Boxplots of DAG from untargeted lipid classes separated by groups. Significant stars are displayed for each comparison with *FDR <0.05, **FDR <0.01, ***FDR <0.001 (limma). (**B**) Boxplots of TAG from untargeted lipid classes separated by groups. (**C**) PCA plot of samples after prior standardization based on significant metabolites between at least one pairwise comparison (limma, FDR <0.05). Variance proportions are written on each component axis. Samples are colored by condition. (**D**) Circular heatmap of the top 159 metabolites (FDR <0.005). Metabolites are represented as slices and labeled around the plot. LogFold Change from significant metabolites between groups is displayed in the first six outer layers. The 7th to 9th layers represents the coefficient of correlation between metabolites and BMI, metabolites and age (Spearman, p <0.1, absolute $R$>0.15) and the p-value from significant associations between metabolites and gender ($\chi^2$, p<0.1).

*Figure 2 continued on next page*

*Figure 2 continued*

The inner layer represents the pathway of each metabolite. (**E**) PCA plot based on metabolites differing clusters adjusted for transmission mode and CD4 count.

The online version of this article includes the following source data and figure supplement(s) for figure 2:

**Source data 1.** Table of differential lipid abundance analysis SNFs by lipids classes by clusters and corrected for transmission mode and CD4 count.

**Figure supplement 1.** Boxplots of untargeted lipid classes separated by groups.

number of MSM than the other groups (***Table 1***). While combining severe and mild at-risk groups, there were 69% (54/78) MSM in the at-risk clusters and 47% (9/19) MSM in the HC-like group. This indicated that sexual preferences and the HIV-1 transmission mode relate to compositional differences in fecal microbiota between clusters. The same effect was observed after correction for transmission mode and CD4 T-count, and alpha diversity did not differ between clusters (***Figure 3—source data 1b***). Permutational multivariate analysis of variance (PERMANOVA) at the family level showed that the centroids of the HC-like groups were different from the severe at-risk (FDR <0.001) and mild groups (FDR = 0.0054; ***Figure 3—source data 2***), indicating that there is only a location effect as permutation test for homogeneity of multivariate dispersions was not significant between the clusters (FDR >0.05). No statistical difference was observed between the severe and mild at-risk groups in both tests (FDR = 0.38). The HC-like group was enriched in Bacteroides and Lachnospira, while at-risk groups were enriched in Prevotella, Veillonella, and Succinivibrio (***Figure 3D–E***). These families were also among 54 significantly discriminative features between HC-like and at-risk groups, as shown with linear discriminant analysis effect size (LefSe; ***Figure 3F***). Mann Whitney U test between clusters at the family level also found Prevotellaceae and Bacteroidaceae to be statistically distinct between these clusters (FDR <0.05; ***Figure 3G***). Our data thus support the potential role of the Prevotella and Bacteroides in the cluster separation that the sexual preferences could mediate in PWH than the metabolic risk cluster.

## Factor and network analysis indicated the importance of microbiome-associated metabolites

To identify the molecular and clinical factors driving SNF cluster separation at the multi-omic level, we employed the Multi-Omic Factor Analysis (MOFA) tool for the multi-omics integration (***Argelaguet et al., 2018***). After low variance filtering, the MOFA model was built using three views: microbiome with 173 ASVs, metabolome with 676 metabolites, and lipidome with 709 lipids. The model found 15 uncorrelated latent factors (***Figure 4—figure supplement 1***), that is, combinations of features at the multi-omic level. The total variance was explained at 80% by the lipidome, 22% by the metabolome, and 2% by the microbiome, agreeing with the SNF analysis (***Figure 4A***). No factor explained most of the variance in the three views (***Figure 4B***). After, we selected features with the largest weight in each cluster-associated factor (***Figure 4C***). Features with the most importance based on the top 10% of absolute weight were selected in each view, resulting in 396 features (263 lipids, 111 metabolites, and 22 ASVs). A good cluster separation based on hierarchical clustering of Spearman correlation confirmed the relevance of this subset of features (***Figure 4D***). We also extracted the top 20 features for each view based on this subset (***Figure 4E***). Bacteroides and Firmicutes were found in the phylum with the highest weight confirming our results from microbiome analysis and the importance of these microbial communities for cluster separation. Nevertheless, the microbiome had a lower weight than metabolites and lipids in MOFA factors. Among the top 20 metabolite features, three metabolites derived or modified by microbiota (defined as microbiome-associated metabolites; MAM) (3,4–dihydroxybutyrate, 2–oxindole–3–acetate, and indoleacetylglutamine) were found (***Figure 4E***). To investigate the coordinated role of MAM, we performed the consensus association analysis (***Figure 4—figure supplement 2***). To balance the different number of features in each of the three omics, we randomly selected 241 metabolites, 241 lipids, and 241 ASVs 1000 times. Significant pairwise correlations (FDR <$10^{-6}$) found in >90% of comparisons were used to build a positive co-expression network, and community detection was performed, resulting in a network with 1324 nodes (694 lipids, 536 metabolites, 94 microbial communities), 131863 edges and eight multi-omic communities (N>30). To refine this network, we selected the 396 features based on MOFA differing the most clusters (***Figure 4D***) in the co-expression network (***Figure 4F***). The most central communities (Average degree C1=444, Average degree C2=364) were lipid specific (SNF-1, lipids = 122/124,

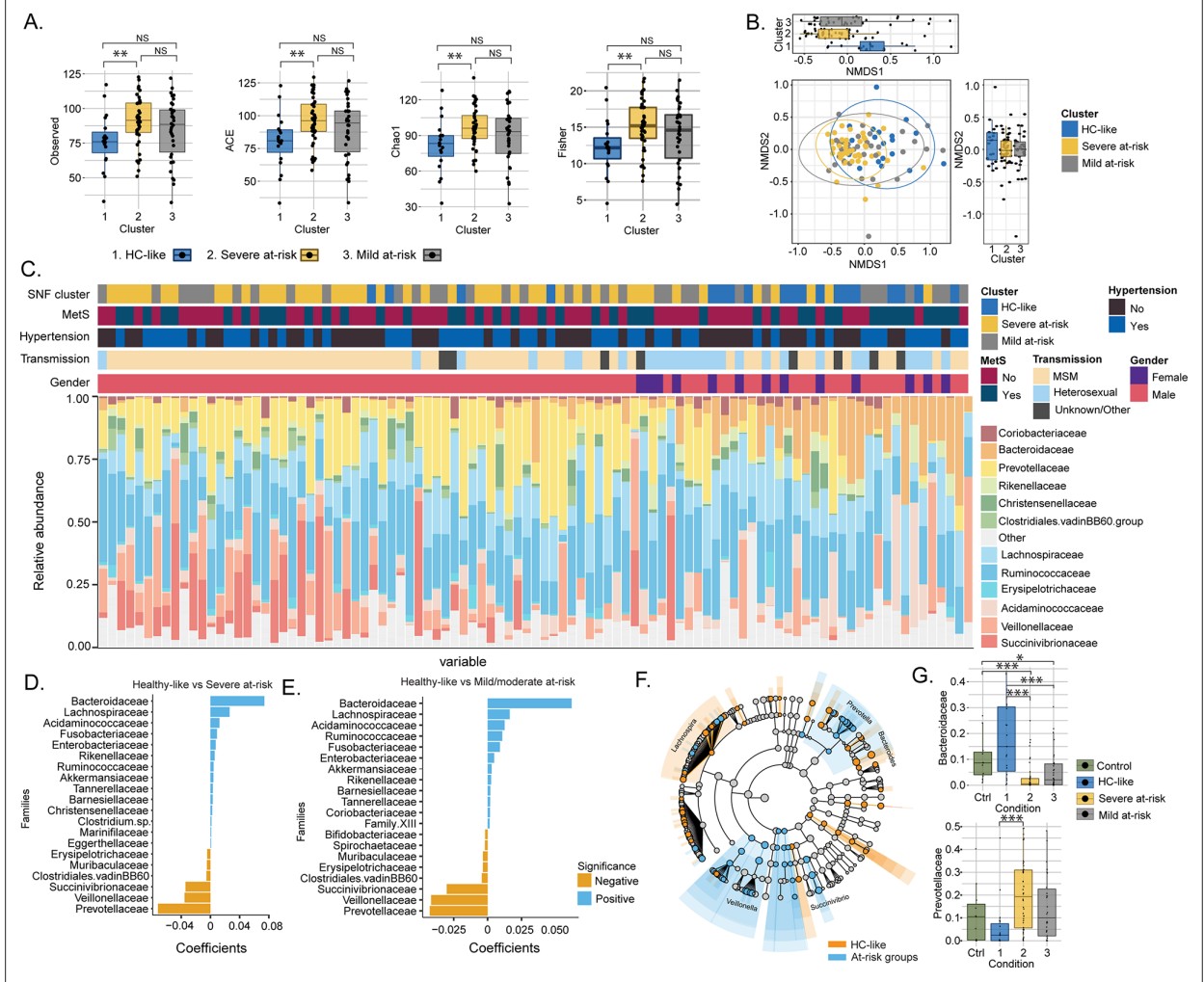

**Figure 3.** Transmission mode drove cluster differences in microbiome data. (**A**) Boxplots of alpha diversity indices (Observed, ACE, Chao1, Fisher) separated by HIV cluster. Significant stars are shown for each comparison (Mann-Whitney U test). (**B**) Non-metric multidimensional scaling (NMDS) plot of Bray-Curtis distances. Samples are colored by clusters. Boxplots based on NMDS1 and NMDS2 are represented. (**C**) Barplot represents the relative abundance of bacteria at the family level for each patient. Patient information is displayed above the barplot, including cluster, metabolic syndrome (MetS: yes/no), hypertension (yes/no), transmission mode, and gender. (**D**) Barplot showing the top microbial families by representing their coefficient from PERMANOVA between SNF-1 and SNF-2. (**E**) Barplot showing the top microbial families between SNF-1 and SNF-3. (**F**) LEfSe cladogram representing cluster-specific microbial communities to HC-like and to at-risk groups (SNF-2/SNF-3). Top families from PERMANOVA are labeled. (**G**) Boxplot of relative abundance at family level for Bacteroides (top) and Prevotella (bottom). Significant stars are shown for significant comparisons (Mann-Whitney U test).

The online version of this article includes the following source data and figure supplement(s) for figure 3:

**Source data 1.** Alpha diversity indices statistics.

**Source data 2.** Permutational multivariate analysis of variance at the family level.

**Figure supplement 1.** Boxplots of alpha diversity indices (se.chao1,Simpson, Shannon, se.ACE, InvSimpson) separated by HIV-cluster.

**Figure supplement 2.** Non-metric multidimensional scaling (NMDS) plot of Bray-Curtis distances.

SNF-2, lipids = 127/128). In contrast, metabolites enriched communities were sparser with a lower average degree (C3=26, C4=22, C6=10, C7=6) but still connected to lipids with 86 edges between lipids and metabolites. Microbiome-enriched community (c8) did not correlate with metabolites or lipids. However, eight MAMs were found in the network, mostly in c6 (5/21), showing that MAMs were highly intercorrelated and could have a potential role in shaping the systemic metabolic and lipid profile.

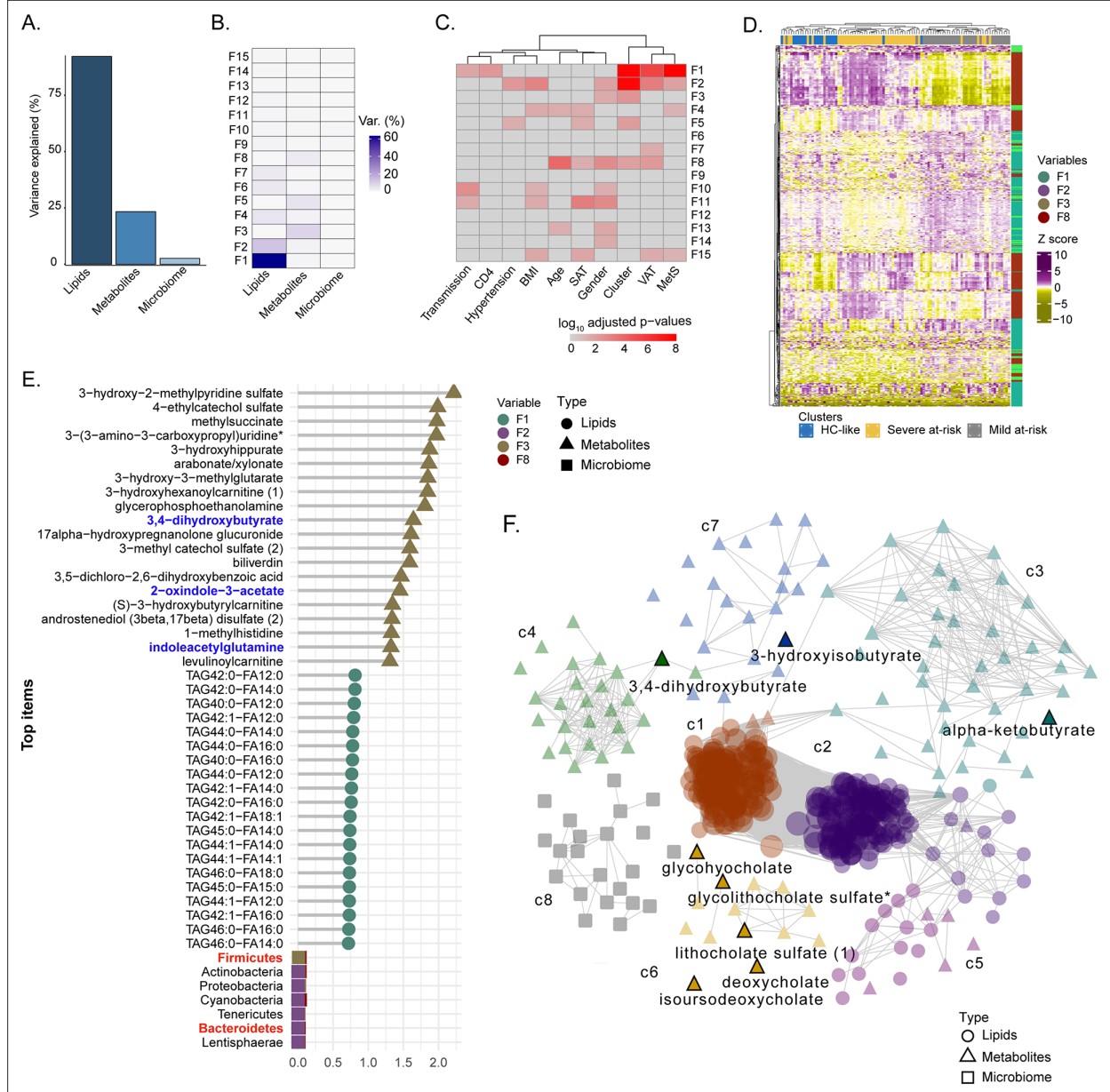

**Figure 4.** Factor analysis highlights the essential features for cluster separation and potential microbiome-derived metabolites importance (**A**) Barplot of total variance explained by MOFA model per view. (**B**) Variance decomposition plot. The percentage of variance is explained by each factor for each view. (**C**) External covariate association with factors plot. Association is represented with log10 adjusted p-values from Pearson correlation. (**D**) Heatmap representing levels of microbial communities, metabolites, and lipids with the higher absolute weight in MOFA factors associated with cluster (**F1, F2, F3, F5, F8**). Samples are labeled according to the study groups. Data were Z-score transformed. The type of data (lipid, metabolite, microbe) is displayed on the right. (**E**) Top 20 features with higher absolute weight in MOFA factors associated with cluster (**F1, F2, F3, F5, F8**) from lipidome, metabolome, and microbiome. Microbiome-derived metabolites and bacterial phylum of interest are colored in blue and red, respectively. (**F**) MOFA features differing clusters and interactions extracted from the three-layers consensus co-expression network. Microbiome-derived metabolites are labeled.

The online version of this article includes the following figure supplement(s) for figure 4:

**Figure supplement 1.** Correlation matrix of MOFA factors.

**Figure supplement 2.** Cytoscape consensus co-expression network.

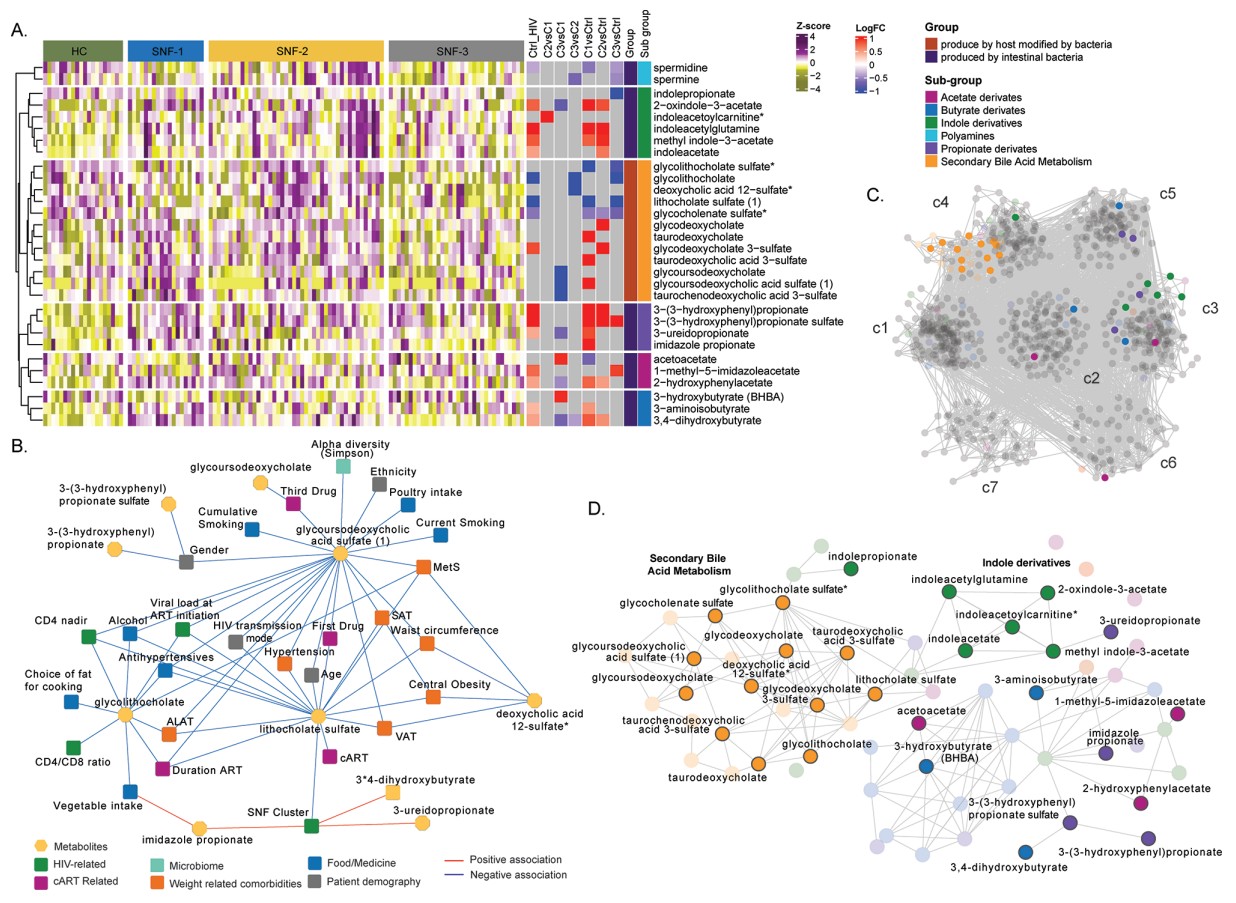

**Figure 5.** Microbiome-associated metabolites are affected in HIV clusters (**A**) Heatmap representing abundances of microbiome-derived metabolites differing in at least one comparison. Data were Z-score transformed. Significant logFC (limma, FDR <0.05) of pairwise comparisons between conditions, groups, and under groups of microbiome-derived metabolites are displayed on the right. (**B**) Cytoscape network showing significant positive and negative associations between clinical parameters and microbiome-derived metabolites (univariate linear regression, FDR <0.05). Clinical parameters are colored based on categories. (**C**) Co-expression network of metabolomics data in PWH. Metabolites are grouped by communities, and microbiome-derived metabolites are labeled and colored based on the subgroup. (**D**) The subset of microbiome-derived metabolites from the co-expression network. Non-significant metabolites in all comparisons are displayed with transparency. Significant microbiome-derived metabolites between at least two conditions are labeled.

The online version of this article includes the following source data for figure 5:

**Source data 1.** Univariate linear regression between clinical parameters and microbiome-derived metabolites differing groups.

## MAM is highly associated with clinical features driven by bile acid metabolism and indole derivatives

We observed a high correlation among the MAMs (*Figure 4F*). Therefore, to further investigate their role in PWH, we retrieved 69 metabolites defined as (i) produced by intestinal bacterial mainly part of secondary bile acid metabolism (n=22) and (ii) produced by host modified by bacteria (n=47, polyamines, propionate, acetate, butyrate, and indole derivatives) as reported (**Appendix 2;** *Postler and Ghosh, 2017*). Differential abundance analysis 19 MAMs differed between HC and PWH irrespective of the SNF clusters, and 30 differed between at least one comparison (*Figure 5A*). The propionate and indole derivates were significantly (FDR <0.05) increased in PWH compared to HC. As observed in the whole metabolomics profile, mild had a more similar profile to HC than HC-like and severe at-risk groups. In contrast, the HC-like and the severe at-risk groups had identical profiles. We performed univariate linear regression to investigate the link between microbiome-derived metabolites and clinical parameters (*Figure 5—source data 1*). Lithocholate sulfate was associated with obesity-related comorbidities (MetS, SAT, VAT, hypertension, and central obesity) and deoxycholic acid 12-sulfate. Several lifestyle parameters impacted MAM, such as poultry and vegetable intake, smoking, and

alcohol. The use of medication as antihypertensives was also associated with three MDMs. Glyco-lithocholate and glycoursodeoxycholic acid sulfate were linked to HIV-related parameters (CD4 nadir, CD4 at study entry) and patients' demography and lifestyle parameters. The SNF cluster was linked to lithocholate sulfate, 3-ureidopropionate, and imidazole propionate (*Figure 5B*). Finally, to measure the influence of MAM on plasma metabolomics profile, we only performed association analysis and community detection on metabolomics data (*Figure 5C*). We obtained a co-expression network with 843 nodes and 15490 edges (FDR <0.02) and observed seven communities (c1-c7) (*Figure 4C*). The c4 contained all the secondary bile acid metabolites. Though the differential abundance analysis did not show all MAM differences between the SNF clusters and HC, they were highly correlated in PWH, with significant MDMs differing between the groups (*Figure 5D*). Combining all the data, we showed the essential role of MAMs in the system-level metabolic profile of PWH on successful therapy.

## Discussion

In this study, we used network and factorization-based integrative analysis of plasma metabolomics, lipidomics, and microbiome profile to characterize clinical phenotypes in the PWH. We identified three different diseases' state-omics phenotypes (HC-like, mild, and severe at-risk) within PWH driven by metabolomics, lipidomics, and microbiome that a single omics or clinical feature could not explain. The integrative omics highlighted the importance of highly intercorrelated microbiome-derived metabolites and their association with the clinical parameters in PWH cluster separation, shaping their systemic health profile. The severe at-risk group (SNF-2) has the at-risk metabolic profile characterized by an increase in TAG and DAG, highest median BMI, MetS incidence, VAT, and SAT, but had a higher CD4 T-cell count at sample collection compared to HC-like and mild at-risk group, which displayed an HC like lipidomic profile. However, the HC-like and severe at-risk group had a similar metabolic profile differing from HC, with dysregulation of AA metabolism. At the microbiome profile, the HC-like group had a lower α-diversity, a lower proportion of MSM, and was enriched in Bacteroides. In contrast, in at-risk groups, there was an increase in Prevotella, with a high proportion of MSM confirming the influence of sexual orientation on the microbiome profile (*Noguera-Julian et al., 2016*). Our study thus identified a risk group of PWH with successful treatment with a dysregulated metabolic profile potentiate metabolic diseases that could be barriers to healthy aging.

Similarity network analysis reduces the high-dimensional nature and different variances of multi-omics data to group patients based on the most similar profile (*Wang et al., 2014*). One of the main advantages of this method is the possibility to compare the networks' similarities to find out which layer has the most similarity with the final network. The similarity network fusion-based patient stratification has been used primarily in non-communicable diseases like cancer [to identify cancer subtypes (*Wang et al., 2014*; *Chierici et al., 2020*) and prognosis (*Wang et al., 2021*)], respiratory diseases (*Narayana et al., 2021*), and to study the influence of diet on human health (*Burton-Pimentel et al., 2021*). Recently we developed SNF-based patient stratification by integrating transcriptomics and metabolomics to define disease severity in COVID-19 that are predictive of the most robust biological features (*Ambikan et al., 2022*). We also reported the influence of gut microbiota on the systemic metabolic profile associated with disease severity (*Albrich et al., 2022*). However, no data were presented to stratify the PWH to fingerprint their disease status. The SNF has shown that the most crucial omics layer in cluster separation was lipids (NMI = 0.6), supported by the MOFA analysis. A study reported that ART and HIV reservoirs are responsible for changes in adipose tissue and lipids metabolism in PWH (*Lagathu et al., 2019*). Dyslipidemia represents the increase in triglycerides, low-density lipoprotein cholesterol (LDL-C), total cholesterol (TC), and decrease of high-density lipoprotein cholesterol (HDL-C) cholesterol in the blood is a well-recognized complication observed in PWH; both naïve (*Wang et al., 2016*) and after ART initiation leading to cardiovascular diseases and mortality (*Bowman and Funderburg, 2019*; *Fiseha et al., 2021*). We found that the severe at-risk individuals (44/97) had most lipids classes upregulated, especially TAG, DAG, and CER, compared to the other groups, while HC-like and mild at-risk groups had no difference with HC. The severe at-risk group also has more patients with high BMI, VAT, SAT, and incidence of MetS. DAG and TAG high levels have been linked to cardiovascular events (*Bowman and Funderburg, 2019*; *Stegemann et al., 2014*). The TAG levels have been linked to insulin resistance and increased diabetes risk (*Bowman and Funderburg, 2019*), confirming this cluster group's qualification as patients with dysregulated lipid profiles and metabolic disease risk. The association of lipid profiles with CD4 counts is still debated. It is positively associated

with (*Fiseha et al., 2021*; *Ji et al., 2019*), and negatively (*Ombeni and Kamuhabwa, 2016*) associated with the highly abundant lipid profile. Interestingly, we found the severe at-risk group to have the highest CD4 count and suppressed viremia but have dysregulated lipid profiles that could be reasoned for unhealthy aging and adverse cardio-metabolic health. Therefore, we propose using a holistic view to define the clinical and immunological treatment success of PWH beyond viral suppression and immune reconstitution.

The second omics-defining clusters were metabolites (NMI = 0.4). Interestingly, the metabolic profile did not completely overlap with the lipid profile showing the complexity associated with the disease. PWH in the HC-like group differed most from the HC regarding their HC-like clinical parameter, with the lowest BMI, VAT, and SAT. Nevertheless, 32% of PWH in the HC-like group had MetS, half of the severe at-risk group (70%) but double the mild at-risk group (17%), indicating a possible lipid-independent metabolic dysregulation. Still, the mild at-risk group had the profile of the most HC-like, similar to the lipids, despite having a significantly higher number of patients with hypertension than the HC-like group. The HC-like and severe at-risk groups showed an up-regulation of the metabolites in the xenobiotics, nucleotides, and AA metabolism, indicating a potential role of diet. We previously showed that the glutamate metabolism was highly disrupted in PWH with MetS in the same COCOMO cohort (*Gelpi et al., 2021*), which can be responsible for late immune recovery in short-term ART patients (*Rosado-Sánchez et al., 2019*). Also, short-chain dicarboxylacylcarnitines (SCDA) and glutamine/valine were higher in PWH with coronary artery disease than in controls (*Okeke et al., 2018*). In our cohort, we observed glutamate, N-acetyl-glutamate, phenyl-acetyl-glutamate, gamma-glutamylglutamate, and 4-hydroxyglutamate part of the glutamine/glutamate metabolism had higher abundance in severe at-risk groups than the mild at-risk group. N-acetyl-glutamate was increased in the mild at-risk group compared to the HC-like group.

The microbiome network had a modest similarity with the final SNF network (NMI = 0.3), and the PCA plot did not observe apparent clustering of patients. Metabolism and immunity of the host are affected by bacteria and disrupted microbiomes linked to illness (*Sun et al., 2016*). More importantly, there is a high variability of microbiota among individuals based on lifestyle, diet, medication, and physiology (*Knight et al., 2018*). Increased α-diversity is associated with good health and decreased diversity in several diseases, including HIV (*Zhou et al., 2020*). A meta-analysis reported that HIV status was not associated with decreased a-diversity in MSM, perhaps due to sexual behaviors, but was decreased in PWH with heterosexual transmission (*Tuddenham et al., 2020*). Despite having healthy clinical and metabolic profiles, we observed a higher α-diversity in the severe at-risk group compared to the HC-like group, probably driven by a higher prevalence of MSM. In terms of bacterial composition, early studies reported that PWH had a higher abundance of Prevotella and a lower abundance of Bacteroides (*Neff et al., 2018*), which in subsequent studies were found to be more related to MSM behaviors than HIV status (*Zhou et al., 2020*; *Gelpi et al., 2020*; *Noguera-Julian et al., 2016*; *Vujkovic-Cvijin et al., 2020*). Our study observed that the severe at-risk group was enriched in Prevotella and depleted in Bacteriodes compared to the HC-like group. Interestingly, the decrease of Bacteroides in obese patients was inversely correlated with serum glutamate (*Wu et al., 2021*), which was also observed in severe at-risk group patients. On the other hand, some Prevotella species have pro-inflammatory effects, leading to intestinal inflammation, bacterial translocation, and microbiome dysbiosis (*Iljazovic et al., 2021*). In general, the complete cohort is mainly composed of MSM (65%, 63/97). As described above, it confirmed that the difference in the microbiome is driven by MSM status in severe at-risk groups, as there was 81% of MSM in that group. The mild at-risk group, even if there is no difference from the severe at-risk group according to PERMANOVA, has the same proportion of MSM as the HC-like group. It has been proposed that early regulation of the MSM-related microbiome could help prevent HIV infection (*Zhou et al., 2020*). However, the question remains whether the MSM-related microbiome is a potential driving force of metabolic comorbidities or whether MSM is a confounding factor disturbing a potentially clinical signal from a disturbed microbiome. Moreover, an increase in Prevotella could potentially aggravate intestinal and systemic inflammation leading to an increased cardiometabolic risk profile (*Iljazovic et al., 2021*; *Littlefield et al., 2022*).

Microbial compositions have implications for metabolism and metabolic diseases, notably through the production of MAMs (*Agus et al., 2021*). Secondary bile acids transformed from primary bile acids by bacteria have a role in lipid digestion. It regulates host metabolism through signaling and

can inhibit the production of pro-inflammatory cytokines by immune cells (*Postler and Ghosh, 2017*). Lipid metabolism, including triglyceride trafficking, is influenced by bile acids through the interaction with the Farnesoid X receptor (FXR) receptor and has been implicated in mice's metabolic disorders (*Schoeler and Caesar, 2019*). A bile acid, glycolithocholate was downregulated in PWH compared to controls previously associated with insulin resistance (*Diboun et al., 2021*). It was negatively associated with food elements such as vegetable intake and choice of fat for cooking, alcohol, and HIV-related parameters such as CD4 levels (nadir and at ART initiation) and HIV duration. High glycodeoxycholate was observed in the at-risk group compared to controls, while glycodeoxycholic acid is negatively associated with insulin resistance (*Wu et al., 2021*). Glycocholenate sulfate was downregulated in the three clusters compared to controls. All secondary bile acids were shown to be highly intercorrelated in co-expression analysis. Three other bile acids, lithocholate sulfate, glycousodesoxycholic acid sulfate, and deoxycholic acid 12-sulfate, were negatively associated with metabolic perturbations, including MetS, VAT, and central obesity. Acetate, propionates, and butyrate are part of short-chain fatty acids (SCFAs) and are obtained from the fiber bacterial fermentation in the colon that the host's enzymes cannot digest (*Alwin and Karst, 2021*). Propionate derivates were upregulated in HC-like and severe at-risk groups. Acetate and butyrate derivates had a more variable profile. Imidazole propionate (IMP) and 3-ureidopropionate were linked to the SNF clusters. In our study, the IMP was also linked to vegetable intake, reportedly involved in insulin resistance (*Agus et al., 2021*). The Bacteroides metabolize most of the acetate and propionate from polysaccharides, and Firmicutes produce butyrate (*Postler and Ghosh, 2017*), which does not explain the relationship within the SNF clusters indicating a more complex interplay between the MAMs and bacterial community in a diseased condition. Tryptophan is converted by bacterial tryptophanase into indole, and indole derivates are involved in the host-microbiota homeostasis (*Krautkramer et al., 2021*). Indoles derivates were mainly upregulated in the HC-like and severe at-risk groups. Our data thus suggested the role of MDMs in shaping the clinical phenotype and systemic health profile in PWH, which could be a therapeutic target for improving health.

Although our study is the first to demonstrate an integrative multi-omics approach to the role of MAMs in systemic alterations in PWH, our study has limitations that merit comments. First, the study is cross-sectional and therefore restricted to predicting dynamic interactions of different omics layers. Second, the microbiome data analysis was done through 16 S methodologies and has a high level of missing data at the genus and species level. Third, although the network-based analysis and the observational data suggest a potential causal association of altered metabolic profile with clinical features, other factors may drive observed effects. Fourth, although this is the largest study to date to perform integrative omics in PWH, the number of samples was relatively low. Finally, microbiome and metabolomics are highly dependent upon an individual's genetics, environment, and diet. The interaction noted may characterize the epiphenomena of a personalized immune system that can be an avenue for future studies to develop a more personalized model for integrative omics to phenotype the disease states we recently reported (*Ambikan et al., 2022*).

In conclusion, we performed a multi-omics analysis of PWH with different clinical features. We identified the diversity of PWH in HIV-related biological alterations regardless of immunological recovery and virological suppression. A proportion of PWH (severe at-risk group around 45% in the present cohort) showed highly dysregulated lipidomics (increased TAG and DAG) and clinical profile (increased BMI and obesity-related features) with increased Prevotella and decreased Bacteroides, the latter being related to MSM transmission. However, alterations in the metabolomics profile and higher CD4 T-cell count at the time of sample collection indicate a complex systemic interplay between host immunity and metabolic health. It can lead to an aggravated higher inflammation profile leading to a cardiometabolic risk profile among the MSM that might affect healthy aging in this population. Integrative analytical approaches that reflect the overall systemic health profile of PWH may improve patient stratification and individual therapeutic and preventive strategies. Given the complex interplay between the clinical and molecular metabolic profile, the application of the multi-omics data for much larger cohorts of PWH might facilitate a better identification of network perturbations and molecular network connections to detect early disease transition toward metabolic complications at an earlier stage. Developing a more personalized model or targeting the interaction networks rather than individual clinical or omics features may provide novel treatment strategies in countering dysregulated metabolic traits, aiming to achieve healthier aging.

## Materials and methods

### Patient cohort and multi-omics data

The cohort comprises 97 PWH from the Copenhagen Comorbidity (COCOMO) Cohort, a prospective cohort of PWH. We used untargeted metabolomics (*Gelpi et al., 2021*), a complex lipid profile (*Olund Villumsen et al., 2021*), and 16 S rRNA microbiome data (*Gelpi et al., 2020*) reported earlier for the larger cohorts. We also extracted clinical and demographic data from the COCOMO database. The HIV-negative controls (HC) (n=20) were used to understand the basal level of omics. Briefly, untargeted metabolomics, which detects the hydrophilic polar compounds, was performed using the Metabolon HD4 Discovery platform (Metabolon Inc, Morrisville, NC 27560, USA) using ultrahigh-performance liquid chromatography/mass spectrometry/mass spectrometry (UHPLC/MS/MS). Untargeted lipidomic was performed through the Complex Lipid Panel technique (Metabolon Inc, Morrisville, NC 27560, USA). The lipid panel covered lipid panels cover Ceramide (CER), Cholesteryl Esters (CE), Diacylglycerols (DAG), Dihydroceramide (DCER), Hexosylceramide (HCER), Lactosylceramide (LCER), Lysophosphatidylcholine (LPC), Lysophosphatidylethanolamine (LPE), Monoacylglycerol (MAG), Phosphatidylcholine (PC), Phosphatidylethanolamine (PE), Phosphatidylinositol (PI), Sphingomyelin (SM), and Triacylglycerols (TAG).

### Omics-driven PWH stratification using Similarity network fusion (SNF)

To stratify the PWH into omics-driven clusters, we used the package SNFtool (*Wang et al., 2014*). Lipids and metabolites with low variance (<0.3) were removed from the data. The microbiome, lipidome, and metabolome were standard normalized before analysis. Pairwise sample distances were calculated with the function dist2 followed by the construction of similarity graphs (number of neighbors, K=13, hyperparameter, alpha = 0.8) for each layer. The similarity network fusion (SNF) was used to all the networks (K=13, number of iterations, T=10) into one. Spectral clustering was applied to the fused network to determine the optimal number of clusters (C=3). The parameters (K, alpha, T, C) were chosen to maximize the Eigengap and minimize rotation cost. The concordance matrix was calculated based on network similarity and measured in normalized mutual information (NMI).

### Lipidomics and metabolomics analysis

Untargeted metabolomics and lipidomics were log2 transformed before analysis. Individual lipid data were grouped by lipid classes as in the following.

$$\left[Class_j\right] = \sum_{i=1}^{n} \left[species_i\right]$$

$$\left[Class_j\right] = \text{Concentration of the lipid class j}$$

$$\left[species_j\right] = \text{Concentration of the molecular species i}$$

$$n = \text{number of molecular species of a class j}$$

The differential abundance analysis was performed pairwise with the R package limma between groups (HC, SNF-1, SNF-2, SNF-3) for lipidomics and metabolomics in two models, one with only clusters and one with clusters, and corrected for factors that differ between the clusters. Benjamini-Hochberg (BH) adjustment was applied.

### Microbiome analysis

Microbiome data analysis was performed using the R package phyloseq (*McMurdie and Holmes, 2013*). The alpha diversity estimates were calculated using the estimate_richness function and the following measures: Observed, ACE, se.ACE, Chao1, Shannon, Simpson, InvSimpson, and Fisher. NMDS ordinations based on Bray-Curtis distances between all samples were calculated using the ordinate function. The vegan package (*Jari Oksanen et al., 2022*) was used to perform PERMANOVA. Equal multivariate dispersion was verified using the betadisper function applying Marti Anderson's PERMDISP2 procedure. Pairwise PERMANOVA test was done between groups using the adonis function, Bray distance, and Bonferroni correction. The cutoff for the adjusted p-value was set up to 0.05. Galaxy module LDA Effect Size (LEfSe) was used to find microbial communities (at genus, family, or higher level) specific to one specific cluster (*Segata et al., 2011*). The multiclass analysis approach was one against all. First, a non-parametric factorial Kruskal-Wallis (KW) sum-rank test was performed with

clusters (cutoff alpha = 0.05), followed by pairwise Wilcoxon rank-sum tests between clusters (cutoff alpha = 0.05), and then effect size calculation for each significant feature was done using discriminant analysis (absolute LDA score >2). Results are represented using a cladogram produced by the module.

## Microbiome-associated metabolites

Microbiome-associated metabolites (MAM), groups, and subgroups were retrieved from the previous literature (*Postler and Ghosh, 2017*) to determine the impact of the microbiome on the metabolism. Univariate linear regression was performed with the function lm between microbiome-derived metabolites and clinical parameters to see the influence of lifestyle on these metabolites.

## Multi-omics factor analysis (MOFA)

MOFA was used to determine the weight of each data type and individual features in PWH. Filtered data for SNF was also used for MOFA analysis (*Argelaguet et al., 2018*). Microbiome data were rarefied by filtering based on variance (>0.2). In addition, the microbiome data were center log-ratio (CLR) transformed to follow a normal distribution. The MOFA model was trained using default parameters, and sample metadata was added to the model. The total variance explained per view was used to see the weight of each omics layer. A correlation plot was used to verify the low correlation between factors. A variance decomposition plot was used to determine the percentage of variance explained by each factor and omics layer. Association analysis of the factors with clinical features was done using the MOFA function correlate_factors_with_covariates and factors associated with the SNF cluster selected. Five and 95% quantile weights for each view were selected for each factor. Pathway analysis was performed on factors using the MOFA function run_enrichment for each view, with the parametric statistical test, FDR-adjusted p-values, and separated positive and negative values. Annotation libraries were made from Metabolon super pathways for metabolomics and lipidomics and Division level for the microbiome.

## Co-expression analysis

We used co-expression analysis to measure the interactions between all features in the data. Pairwise Spearman correlations between features were calculated using the R package stat, and the cutoff for FDR of significant correlations was selected to minimize the number of false positives. The positive and negative networks were built using the python igraph (*Csárdi and Nepusz, 2005*) and compared to random networks of the same size. Leiden community detection was applied to find groups of interconnected features, and the mean degree was calculated to represent the community centrality using the python module leidenag (*Blondel et al., 2008*). Communities of less than 30 features were excluded. Consensus association analysis was performed to integrate the three layers of omics using 1000 iterations. At each iteration, pairwise correlations between ASVs (N=241), 241 metabolites, and 241 lipids selected randomly were run, and significant positive correlations (Spearman, FDR <0.001) were kept as an association. Associations found in 90% of the comparisons over all iterations were kept building the final network as described above.

## General statistics

Differences between clusters in clinical parameters were measured using Kruskal–Wallis H test for continuous variables and Chi-Square Test or Fisher's Exact Test for discrete variables. Deviations were mentioned in all respective analyses. The default p-value cutoff was set to 0.05. Other p-values cutoffs are adapted for a specific analysis depending upon the number of significance and to minimize the false positivity (*Team TRDC, 2010*).

## Visualization

Scatter plots, PCA plots, box plots, NMDS plots, circular heatmap, and bar plots were generated using ggplot2 (*Wickham, 2016*). Heatmaps were generated using ComplexHeatmap (*Gu et al., 2016*). Sankey plot was made using the R package ggalluvial (*Brunson, 2020*). Networks were plotted using Cytoscape v3.6.1 (*Shannon et al., 2003*).

## Acknowledgements

The study is funded by the Swedish Research Council grants 2017–01330, 2018–06156, and 2021–01756 to UN Novo Nordic Foundation, Lundbeck Foundation, Augustinus Foundation, Region Hovedstaden, and Rigshospitalet Research Council to SDN. DDM acknowledges the support received from Danish National Research Foundation Grant 126 (DNRF126).

## Additional information

### Funding

| Funder | Grant reference number | Author |
|---|---|---|
| Vetenskapsrådet | 2017-01330 | Ujjwal Neogi |
| Novo Nordisk | | Susanne D Nielsen |
| Vetenskapsrådet | 2018-06156 | Ujjwal Neogi |
| Vetenskapsrådet | 2021-01756 | Ujjwal Neogi |
| Danmarks Grundforskningsfond | 126 (DNRF126) | Daniel Murray |
| Lundbeck Foundation | | Susanne D Nielsen |
| Augustinus Foundation | | Susanne D Nielsen |
| Region Hovedstaden | | Susanne D Nielsen |
| Rigshospitalet | | Susanne D Nielsen |

The funders had no role in study design, data collection and interpretation, or the decision to submit the work for publication.

### Author contributions

Flora Mikaeloff, Data curation, Formal analysis, Visualization, Writing - original draft; Marco Gelpi, Data curation, Formal analysis, Investigation, Writing – review and editing; Rui Benfeitas, Software, Supervision, Investigation, Project administration, Writing – review and editing; Andreas D Knudsen, Data curation, Investigation, Methodology; Beate Vestad, Formal analysis, Methodology, Writing – review and editing; Julie Høgh, Johannes R Hov, Data curation, Formal analysis, Writing – review and editing; Thomas Benfield, Data curation, Investigation, Writing – review and editing; Daniel Murray, Data curation, Funding acquisition, Investigation, Writing – review and editing; Christian G Giske, Supervision, Writing – review and editing; Adil Mardinoglu, Supervision, Funding acquisition, Methodology, Writing – review and editing; Marius Trøseid, Data curation, Formal analysis, Supervision, Methodology, Writing – review and editing; Susanne D Nielsen, Conceptualization, Resources, Investigation, Methodology, Project administration, Writing – review and editing; Ujjwal Neogi, Conceptualization, Resources, Supervision, Validation, Visualization, Methodology, Writing - original draft, Project administration

### Author ORCIDs

Flora Mikaeloff ⓘ http://orcid.org/0000-0002-6171-7043
Thomas Benfield ⓘ http://orcid.org/0000-0003-0698-9385
Ujjwal Neogi ⓘ http://orcid.org/0000-0002-0844-3338

### Ethics

Human subjects: Ethical approval was obtained by the Regional Ethics Committee of Copenhagen (COCOMO: H-15017350) and Etikprövningsmyndigheten, Sweden (Dnr: 2022-01353-01). Informed consent was obtained from all participants and delinked before analysis.

### Decision letter and Author response

Decision letter https://doi.org/10.7554/eLife.82785.sa1
Author response https://doi.org/10.7554/eLife.82785.sa2

## Additional files

### Supplementary files
• Supplementary file 1. Table of differential lipid abundance analysis on individual lipids abundances between clusters corrected for transmission mode and CD4 count.
• Supplementary file 2. False discovery rate after differential metabolite analysis between the groups.
• Supplementary file 3. Table of differential metabolite abundance analysis after adjustment of the CD4 count at sampling and route of transmission.
• MDAR checklist

### Data availability
All of the data generated or analyzed during this study are included in this published article and/or the supplementary materials. Created datasets and code are publicly available. The metabolomics and lipidomics data are available from https://doi.org/10.6084/m9.figshare.14356754.v1 and https://doi.org/10.6084/m9.figshare.14509452.v1. All the codes are available at github: https://github.com/neogilab/HIV_multiomics, (copy archived at swh:1:rev:86aae862497b7dbb3dae4ce2e5a44b0369e0dec0).

The following datasets were generated:

| Author(s) | Year | Dataset title | Dataset URL | Database and Identifier |
|---|---|---|---|---|
| Neogi U, Nielsen SD | 2022 | Original Scale Metabolomics data: COCOMO | https://doi.org/10.6084/m9.figshare.14356754.v1 | figshare, 10.6084/m9.figshare.14356754.v1 |
| Neogi U, Nielsen SD | 2022 | Original Scale Data: COCOMO_Lipidomics | https://doi.org/10.6084/m9.figshare.14509452.v1 | figshare, 10.6084/m9.figshare.14509452.v1 |

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

# Appendix 1

**Appendix 1—table 1.** List of parameters used in the study.

| Description | Type | Definition |
|---|---|---|
| Ethnicity defined as in the Danish HIV cohort | logical | 1: caucasian, 2: asian, 3: black, 4: other 5: inuit |
| Origin defined as in the Danish HIV cohort | logical | 1: Denmark, 2: Other Scandinavian country 3: Other European country, 4: Turkia, 5: Pakistan, India and Sri Lanka, 6: Arabian countries / Iran, 7: Other (Africa, Asia, Greenland) |
| Mode of HIV transmission | logical | 1.Homo/biseksuel, 2. IV drug use, 3.homoseksuel+misbrug4.hæmofili 5.blodtransfusion 6.heteroseksuel, 7. andet 8. unknown 9.perinatal |
| Gender | logical | 0: female, 1:male |
| Computed age | numeric | Unit: years |
| Body mass index | numeric | Unit: kg/m$^2$ |
| Physical activity in spare time | logical | 1: Inactive 2: Slightly active 3: Moderately active 4: Very active |
| Meat intake (Beef) | numeric | Times per week in average |
| Meat intake (Poultry) | numeric | Times per week in average |
| Choice of fat for preparing warm dishes | logical | 0: None, 1: Butter, 2: Spreadable (*Kærgården)*, 3: Shortening (*Stegemargarine),* 4: Vegetable margarine (*Plantemargarine),* 5: Olive oil, 6: Other oil, 7: Other |
| Vegetable intake | logical | 0: Never 1: 1–3/month, 2: 1–2/week, 3: 3–4/week, 4: 5–6/week, 5: 1/day, 6: 2–3/day, 7:>3 /day |
| Fruit intake (whole fruit/portion of fruit) | logical | 0: Never, 1: 1–3/month, 2: 1–2/week, 3: 3–4/week, 4: 5–6/week, 5: 1/day, 6: 2–3/day, 7:>3 /day |
| *Weekly alcohol consumption | numeric | Unit:gram |
| Cumulative smoking | numeric | Unit: Pack years in current and previous smokers |
| Current Smoking | logical | 0: No, 1: Yes |
| Current CD4 count (closest to date of inclusion) | numeric | cell / ul |
| Current CD8 count | numeric | cell / ul |
| Ratio CD4/CD8 | numeric | x |
| CD4_nadir | numeric | cell / ul |
| CD4 at ART initiation | numeric | cell / ul |
| Current viral load | numeric | copies / ul |
| VL at ART initiation | numeric | copies / ul |
| Log10 of VL at ART initiation | numeric | x |
| SNF clusters | logical | 1:SNF-1, 2: SNF-2, 3: SNF-3 |
| Duration of current cART | numeric | Unit: months |
| Duration of current cART | numeric | Unit: years |
| Duration of previous cART | numeric | Unit: months |
| 3[rd] Drug | logical | NNRTI, PI/r,INSTI, Other/unknow |
| 1st drug | logical | ABC, TDF/TAF, Other/unknown |
| Subcutaneous adipose tissue (SAT) | numeric | squared centimeters |

*Appendix 1—table 1 Continued on next page*

*Appendix 1—table 1 Continued*

| Description | Type | Definition |
| --- | --- | --- |
| Visceral adipose tissue (VAT) | numeric | squared centimeters |
| Waist circumference | numeric | Unit: cm |
| Systolic blood pressure, right arm | numeric | Unit: mm mercury |
| Diastolic blood pressure, right arm | numeric | Unit: mm mercury |
| Metabolic syndrome (MetS)* | logical | 0: No, 1: Yes |
| Central obesity | logical | 0: No,1: Yes |
| hypertension (*JNC7 definition) | logical | 0: No, 1: Yes |
| Diabetes | logical | 0: No, 1: Yes |
| Estimated glomerular function rate (eGFR) | numeric | Unit: reads mL/min/1.73 m2 |
| Alanine aminotransferase (ALAT) | numeric | Unit:Units per liter |
| Antihypertensives | logical | 0: No, 1: Yes |

# Appendix 2

**Appendix 2—table 1.** List of microbiome-derived metabolites.

| BIOCHEMICAL | SUPER.PATHWAY | SUB.PATHWAY | group | under_group |
|---|---|---|---|---|
| butyrate/isobutyrate (4:0) | Lipid | Short Chain Fatty Acid | produced_by_intestinal_bacteria | Short Chain Fatty Acid |
| N-acetylputrescine | Amino Acid | Polyamine Metabolism | produced_by_intestinal_bacteria | Polyamines |
| spermidine | Amino Acid | Polyamine Metabolism | produced_by_intestinal_bacteria | Polyamines |
| spermine | Amino Acid | Polyamine Metabolism | produced_by_intestinal_bacteria | Polyamines |
| 1-methyl-4-imidazoleacetate | Amino Acid | Histidine Metabolism | produced_by_intestinal_bacteria | Acetate derivates |
| 1-methyl-5-imidazoleacetate | Amino Acid | Histidine Metabolism | produced_by_intestinal_bacteria | Acetate derivates |
| 1-ribosyl-imidazoleacetate* | Amino Acid | Histidine Metabolism | produced_by_intestinal_bacteria | Acetate derivates |
| 1H-indole-7-acetic acid | Xenobiotics | Bacterial/Fungal | produced_by_intestinal_bacteria | Indole derivatives |
| 2,3-dihydroxy-2-methylbutyrate | Amino Acid | Leucine, Isoleucine and Valine Metabolism | produced_by_intestinal_bacteria | Butyrate derivates |
| 2-aminobutyrate | Amino Acid | Glutathione Metabolism | produced_by_intestinal_bacteria | Butyrate derivates |
| 2-hydroxybutyrate/2-hydroxyisobutyrate | Amino Acid | Glutathione Metabolism | produced_by_intestinal_bacteria | Butyrate derivates |
| 2-hydroxyphenylacetate | Amino Acid | Phenylalanine Metabolism | produced_by_intestinal_bacteria | Acetate derivates |
| 2-oxindole-3-acetate | Xenobiotics | Food Component/Plant | produced_by_intestinal_bacteria | Indole derivatives |
| 2 R,3R-dihydroxybutyrate | Lipid | Fatty Acid, Dihydroxy | produced_by_intestinal_bacteria | Butyrate derivates |
| 2 S,3R-dihydroxybutyrate | Lipid | Fatty Acid, Dihydroxy | produced_by_intestinal_bacteria | Butyrate derivates |
| 3,4-dihydroxybutyrate | Lipid | Fatty Acid, Dihydroxy | produced_by_intestinal_bacteria | Butyrate derivates |
| 3-(3-hydroxyphenyl)propionate | Xenobiotics | Benzoate Metabolism | produced_by_intestinal_bacteria | Propionate derivates |
| 3-(3-hydroxyphenyl)propionate sulfate | Xenobiotics | Benzoate Metabolism | produced_by_intestinal_bacteria | Propionate derivates |
| 3-aminoisobutyrate | Nucleotide | Pyrimidine Metabolism, Thymine containing | produced_by_intestinal_bacteria | Butyrate derivates |
| 3-carboxy-4-methyl-5-pentyl-2-furanpropionate (3-CMPFP) | Lipid | Fatty Acid, Dicarboxylate | produced_by_intestinal_bacteria | Propionate derivates |
| 3-formylindole | Xenobiotics | Food Component/Plant | produced_by_intestinal_bacteria | Indole derivatives |
| 3-hydroxy-2-ethylpropionate | Amino Acid | Leucine, Isoleucine and Valine Metabolism | produced_by_intestinal_bacteria | Propionate derivates |
| 3-hydroxybutyrate (BHBA) | Lipid | Ketone Bodies | produced_by_intestinal_bacteria | Butyrate derivates |
| 3-hydroxyisobutyrate | Amino Acid | Leucine, Isoleucine and Valine Metabolism | produced_by_intestinal_bacteria | Butyrate derivates |
| 3-indoleglyoxylic acid | Xenobiotics | Food Component/Plant | produced_by_intestinal_bacteria | Indole derivatives |
| 3-methyl-2-oxobutyrate | Amino Acid | Leucine, Isoleucine and Valine Metabolism | produced_by_intestinal_bacteria | Butyrate derivates |

*Appendix 2—table 1 Continued on next page*

*Appendix 2—table 1 Continued*

| BIOCHEMICAL | SUPER.PATHWAY | SUB.PATHWAY | group | under_group |
|---|---|---|---|---|
| 3-phenylpropionate (hydrocinnamate) | Xenobiotics | Benzoate Metabolism | produced_by_intestinal_bacteria | Propionate derivates |
| 3-ureidopropionate | Nucleotide | Pyrimidine Metabolism, Uracil containing | produced_by_intestinal_bacteria | Propionate derivates |
| 4-imidazoleacetate | Amino Acid | Histidine Metabolism | produced_by_intestinal_bacteria | Acetate derivates |
| 6-hydroxyindole sulfate | Xenobiotics | Chemical | produced_by_intestinal_bacteria | Indole derivatives |
| 7-hydroxyindole sulfate | Amino Acid | Tryptophan Metabolism | produced_by_intestinal_bacteria | Indole derivatives |
| acetoacetate | Lipid | Ketone Bodies | produced_by_intestinal_bacteria | Acetate derivates |
| alpha-ketobutyrate | Amino Acid | Methionine, Cysteine, SAM and Taurine Metabolism | produced_by_intestinal_bacteria | Butyrate derivates |
| citalopram propionate* | Xenobiotics | Drug - Psychoactive | produced_by_intestinal_bacteria | Propionate derivates |
| gamma-glutamyl-2-aminobutyrate | Peptide | Gamma-glutamyl Amino Acid | produced_by_intestinal_bacteria | Butyrate derivates |
| guanidinoacetate | Amino Acid | Creatine Metabolism | produced_by_intestinal_bacteria | Acetate derivates |
| hydantoin-5-propionate | Amino Acid | Histidine Metabolism | produced_by_intestinal_bacteria | Propionate derivates |
| imidazole propionate | Amino Acid | Histidine Metabolism | produced_by_intestinal_bacteria | Propionate derivates |
| iminodiacetate (IDA) | Xenobiotics | Chemical | produced_by_intestinal_bacteria | Acetate derivates |
| indole-3-carboxylate | Amino Acid | Tryptophan Metabolism | produced_by_intestinal_bacteria | Indole derivatives |
| indoleacetate | Amino Acid | Tryptophan Metabolism | produced_by_intestinal_bacteria | Indole derivatives |
| indoleacetoylcarnitine* | Amino Acid | Tryptophan Metabolism | produced_by_intestinal_bacteria | Indole derivatives |
| indoleacetylglutamine | Amino Acid | Tryptophan Metabolism | produced_by_intestinal_bacteria | Indole derivatives |
| indolelactate | Amino Acid | Tryptophan Metabolism | produced_by_intestinal_bacteria | Indole derivatives |
| indolepropionate | Amino Acid | Tryptophan Metabolism | produced_by_intestinal_bacteria | Indole derivatives |
| methyl indole-3-acetate | Xenobiotics | Food Component/Plant | produced_by_intestinal_bacteria | Indole derivatives |
| phenylacetate | Amino Acid | Phenylalanine Metabolism | produced_by_intestinal_bacteria | Acetate derivates |
| taurine | Amino Acid | Methionine, Cysteine, SAM and Taurine Metabolism | produced_by_host_modified_by_bacteria | Taurine |
| deoxycholate | Lipid | Secondary Bile Acid Metabolism | produced_by_host_modified_by_bacteria | Secondary Bile Acid Metabolism |
| deoxycholic acid 12-sulfate* | Lipid | Secondary Bile Acid Metabolism | produced_by_host_modified_by_bacteria | Secondary Bile Acid Metabolism |
| deoxycholic acid glucuronide | Lipid | Secondary Bile Acid Metabolism | produced_by_host_modified_by_bacteria | Secondary Bile Acid Metabolism |
| glycocholenate sulfate* | Lipid | Secondary Bile Acid Metabolism | produced_by_host_modified_by_bacteria | Secondary Bile Acid Metabolism |
| glycodeoxycholate | Lipid | Secondary Bile Acid Metabolism | produced_by_host_modified_by_bacteria | Secondary Bile Acid Metabolism |
| glycodeoxycholate 3-sulfate | Lipid | Secondary Bile Acid Metabolism | produced_by_host_modified_by_bacteria | Secondary Bile Acid Metabolism |

*Appendix 2—table 1 Continued on next page*

*Appendix 2—table 1 Continued*

| BIOCHEMICAL | SUPER.PATHWAY | SUB.PATHWAY | group | under_group |
|---|---|---|---|---|
| glycohyocholate | Lipid | Secondary Bile Acid Metabolism | produced_by_host_modified_by_bacteria | Secondary Bile Acid Metabolism |
| glycolithocholate | Lipid | Secondary Bile Acid Metabolism | produced_by_host_modified_by_bacteria | Secondary Bile Acid Metabolism |
| glycolithocholate sulfate* | Lipid | Secondary Bile Acid Metabolism | produced_by_host_modified_by_bacteria | Secondary Bile Acid Metabolism |
| glycoursodeoxycholate | Lipid | Secondary Bile Acid Metabolism | produced_by_host_modified_by_bacteria | Secondary Bile Acid Metabolism |
| glycoursodeoxycholic acid sulfate (1) | Lipid | Secondary Bile Acid Metabolism | produced_by_host_modified_by_bacteria | Secondary Bile Acid Metabolism |
| hyocholate | Lipid | Secondary Bile Acid Metabolism | produced_by_host_modified_by_bacteria | Secondary Bile Acid Metabolism |
| isoursodeoxycholate | Lipid | Secondary Bile Acid Metabolism | produced_by_host_modified_by_bacteria | Secondary Bile Acid Metabolism |
| lithocholate sulfate (1) | Lipid | Secondary Bile Acid Metabolism | produced_by_host_modified_by_bacteria | Secondary Bile Acid Metabolism |
| taurochenodeoxycholic acid 3-sulfate | Lipid | Secondary Bile Acid Metabolism | produced_by_host_modified_by_bacteria | Secondary Bile Acid Metabolism |
| taurocholenate sulfate* | Lipid | Secondary Bile Acid Metabolism | produced_by_host_modified_by_bacteria | Secondary Bile Acid Metabolism |
| taurodeoxycholate | Lipid | Secondary Bile Acid Metabolism | produced_by_host_modified_by_bacteria | Secondary Bile Acid Metabolism |
| taurodeoxycholic acid 3-sulfate | Lipid | Secondary Bile Acid Metabolism | produced_by_host_modified_by_bacteria | Secondary Bile Acid Metabolism |
| taurohyocholate* | Lipid | Secondary Bile Acid Metabolism | produced_by_host_modified_by_bacteria | Secondary Bile Acid Metabolism |
| taurolithocholate 3-sulfate | Lipid | Secondary Bile Acid Metabolism | produced_by_host_modified_by_bacteria | Secondary Bile Acid Metabolism |
| ursodeoxycholate | Lipid | Secondary Bile Acid Metabolism | produced_by_host_modified_by_bacteria | Secondary Bile Acid Metabolism |

