## [Editor Report]

This important study systematically integrates convincing multi-omics data to identify the metabolic at-risk profiles within people living with HIV on antiretroviral therapy and presents findings that have focused importance and scope. The authors have used appropriate and validated methodology in line with the current state-of-the-art and have produced a paper that is of great interest to a specialised audience interested in HIV infection and metabolic mechanisms.

---

## [Decision Letter]

**Decision letter after peer review:**

Thank you for submitting your article "Network-based multi-omics integration reveals metabolic at-risk profile within treated HIV-infection" for consideration by *eLife*. Your article has been reviewed by 2 peer reviewers, and the evaluation has been overseen by a Reviewing Editor and Jos van der Meer as the Senior Editor. The reviewers have opted to remain anonymous.

Essential revisions:

1) Table 1: The HC group should be added and compared for comparability assessment of characteristics with the HIV patient groups.

2) In addition to the BMI, hypertension, central obesity, etc., it would be very important to assess and report the status of non-communicable chronic comorbidities such as cardiovascular diseases, kidney functions, non-alcoholic fatty liver diseases among these aging participants who had been on long time ART, which are closely related with metabolic and lipidomic alterations. Moreover, it would also be critical to report the alcohol use and dietary habit of the participants, which are usually associated with gut microbiota.

3) When comparing metabolic or lipidomic or microbiome alterations or characteristics between different SNF groups and/or the HC group, as well as trying to establish associations, potential confounding variables should have been controlled by multiple regression analysis. The inability of the study to make appropriate adjustments might explain the controversial findings in terms of various associations or differences/similarities in metabolic alterations versus microbiome and α-diversity between different SNF groups and/or the HC group.

4) It is very difficult to follow the methodology section in terms of the rationale for the methods used. Moreover, reproducibility is currently hampered by a succession of different methods.

*Reviewer #1 (Recommendations for the authors):*

1. Table 1: The HC group should be added and compared for comparability assessment of characteristics with the HIV patient groups.

2. In addition to the BMI, hypertension, central obesity, etc., it would be very important to assess and report the status of non-communicable chronic comorbidities such as cardiovascular diseases, kidney functions, non-alcoholic fatty liver diseases among these aging participants who had been on long time ART, which are closely related with metabolic and lipidomic alterations. Moreover, it would also be critical to report the alcohol use and dietary habit of the participants, which are usually associated with gut microbiota.

3. When comparing metabolic or lipidomic or microbiome alterations or characteristics between different SNF groups and/or the HC group, as well as trying to establish associations, potential confounding variables should have been controlled by multiple regression analysis. The inability of the study to make appropriate adjustments might explain the controversial findings in terms of various associations or differences/similarities in metabolic alterations versus microbiome and α-diversity between different SNF groups and/or the HC group.

4. Not clear if the plasma and fecal samples were tested at the same time and same performance.

5. There are many obvious grammatical errors, editorial mistakes, and typos in the manuscript.

*Reviewer #2 (Recommendations for the authors):*

General comment: What would be the medical/clinical value of identifying metabolic at-risk profiles? Please provide relevant insights in the Abstract, Introduction, and Discussion (or even in the Results). It would be great if this clarification is made with a specific potential scenario in Results or Discussion rather than a general statement.

Abstract: Would it be possible to provide a few more conclusive insights drawn from the multi-omics analysis? The current version provides the results, but it seems that they can be presented in a more integrative/coherent manner.

The legend of Figure 3G was omitted, and not cited in the main text.

The third paragraph in the Discussion section: It would be nice to provide information on metabolic pathways that are associated with some of the relatively unknown, upregulated metabolites, including N-acetyl-glutamate, phenyl-acetylglutamate, γ-glutamylglutamate, and 4-hydroxyglutamate.

Lipidomics is a subset of metabolites. In this study, they were separately considered. In this regard, was any additional data preparation conducted for lipidomics data? If so, please provide relevant information.

---

## [Author Response]

Reviewer #1 (Recommendations for the authors):2. In addition to the BMI, hypertension, central obesity, etc., it would be very important to assess and report the status of non-communicable chronic comorbidities such as cardiovascular diseases, kidney functions, non-alcoholic fatty liver diseases among these aging participants who had been on long time ART, which are closely related with metabolic and lipidomic alterations. Moreover, it would also be critical to report the alcohol use and dietary habit of the participants, which are usually associated with gut microbiota.

We are thankful to the reviewer for the suggestion. As suggested, we have added all the data to the analysis. We added non-communicable chronic comorbidities. There are no differences in ALAT (Liver function), eGFR (kidney function), and diabetes between clusters (all pval>0.05). We also checked the alcohol, smoking, and dietary habits of participants, and we did not observe differences between clusters, and this is why we did not correct them in our analysis. We have now presented the data as source data Table 1-source data 1.

3. When comparing metabolic or lipidomic or microbiome alterations or characteristics between different SNF groups and/or the HC group, as well as trying to establish associations, potential confounding variables should have been controlled by multiple regression analysis. The inability of the study to make appropriate adjustments might explain the controversial findings in terms of various associations or differences/similarities in metabolic alterations versus microbiome and α-diversity between different SNF groups and/or the HC group.

We are thankful for the suggestion. We do not believe our study is controversial as similar studies were not performed on HIV. The outcome of our study is metabolic complications that include high BMI, SAT, and VAT, the incidence of metabolic syndrome, height weight circumference, and hypertension. We identified two confounding variables unrelated to the metabolic complications, i.e., transmission mode and the CD4 count (Table 1). As the reviewer recommended, we corrected our models for transmission mode and CD4 count.

In the metabolomics data (please see Author response image 1), a relatively low difference was observed between the noncorrected and corrected models. However, as suggested by the reviewer, we are now presenting the analysis after correction in supplementary table S3.

**Author response image 1. sa2fig1:** Venn diagram of metabolites significantly different between clusters in noncorrected limma model (left) and model corrected for transmission mode and CD4 count (right).

More, we observed a higher variance explained by the two first components in PCA analysis using metabolites significantly different between clusters and corrected for transmission mode and CD4 count (Figure 2E).

In the lipidomics data, after correction for transmission mode, we observed similar results with TAG and DAG differing between SNF-2 and other clusters (please see Author response image 2). However, as suggested by the reviewer, we are now presenting the analysis after correction in supplementary table S3.

**Author response image 2. sa2fig2:** Venn diagram of lipids significantly different between clusters in noncorrected limma model (left) and model corrected for transmission mode and CD4 count (right).

In the microbiome data, our central hypothesis is that transmission mode has a stronger influence on the microbiome than our potential metabolic-driven profile. Accordingly, after correction for transmission mode and CD4 count, α diversity indices were not significantly different between clusters. The results of corrected and non-corrected models were updated in supplementary table S4.

In the profile of microbiome-associated metabolites, almost no changes were observed, and similar pathways with different clusters were identified. As suggested by the reviewer, we have now updated the figure with Figure 5, and Figure 5-source data 1.

All these changes do not change our results but rather strengthen our findings. We are thankful to the reviewer for the suggestions.

4. Not clear if the plasma and fecal samples were tested at the same time and same performance.

Yes, both the plasma and fecal samples were collected at the same time. It is now mentioned in the method.

5. There are many obvious grammatical errors, editorial mistakes, and typos in the manuscript.

We have now corrected the grammatical errors, editorial mistakes, and typos in the manuscript.

Reviewer #2 (Recommendations for the authors):General comment: What would be the medical/clinical value of identifying metabolic at-risk profiles? Please provide relevant insights in the Abstract, Introduction, and Discussion (or even in the Results). It would be great if this clarification is made with a specific potential scenario in Results or Discussion rather than a general statement.

We are thankful to the reviewer for the suggestion. System biology's application in identifying a disease state's biological mechanism in HIV-infected individuals is a relatively new field. We agree with the reviewer that connecting the findings in this study with specific medical/clinical insights is the next challenge. However, the first proof-of-concept study on 108 patients showed that multi-omics studies could generate a correlation network of communities of related analytes associated with physiology and disease. More importantly, the behavioral coaching informed by personal data helped participants to improve clinical biomarkers [PMID: 28714965]. The applications of multi-omics data are more and more valuable in non-communicable diseases [PMID: 35528975, PMID: 36503356 etc.]. As suggested by the reviewer, we have now elaborated on the medical/clinical value in identifying metabolic at-risk profiles, in particular the potential to improve individual risk stratification and to personalize lifestyle interventions. Still, as our study is an association study, data should be regarded as exploratory, and not sufficient to suggest any changes in clinical practice.

We have concluded the manuscript as follows:

“However, alterations in the metabolomics profile and higher CD4 T-cell count at the time of sample collection indicate a complex systemic interplay between host immunity and metabolic health. It can lead to an aggravated higher inflammation profile leading to a cardiometabolic risk profile among the MSM that might affect healthy aging in this population. Integrative analytical approaches that reflect the overall systemic health profile of PWH may improve patient stratification and individual therapeutic and preventive strategies. Given the complex interplay between the clinical and molecular metabolic profile, the application of the multi-omics data for much larger cohorts of PWH might facilitate a better identification of network perturbations and molecular network connections to detect early disease transition toward metabolic complications at an earlier stage. Developing a more personalized model or targeting the interaction networks rather than individual clinical or omics features may provide novel treatment strategies in countering dysregulated metabolic traits, aiming to achieve healthier aging.”

Abstract: Would it be possible to provide a few more conclusive insights drawn from the multi-omics analysis? The current version provides the results, but it seems that they can be presented in a more integrative/coherent manner.

We are thankful for the suggestion. We have modified as suggested.

The legend of Figure 3G was omitted, and not cited in the main text.

We are thankful to the reviewer for pointing out the mistake. There was no figure 3G because it was redundant with the statistics of the clinical parameters and was not adding additional relevant information for the study.

The third paragraph in the Discussion section: It would be nice to provide information on metabolic pathways that are associated with some of the relatively unknown, upregulated metabolites, including N-acetyl-glutamate, phenyl-acetylglutamate, γ-glutamylglutamate, and 4-hydroxyglutamate.

As suggested, we have added the pathways. They are part of glutamate/glutamine metabolism.

Lipidomics is a subset of metabolites. In this study, they were separately considered. In this regard, was any additional data preparation conducted for lipidomics data? If so, please provide relevant information.

We are thankful for the comments. We are sorry that we did not mention the methodology of metabolomics and lipidomics carefully in the manuscript. Metabolomics mainly focuses on hydrophilic polar compounds, while lipidomics has emerged as independent omics owing to the complexities of the organismal lipidomes and hydrophobic nonpolar compounds. Though metabolomics provides some degree of the compound of lipid metabolism e.g., Bile Acids Bioactive Lipids, Cholesterol, Fatty Acids, Sphingosine, Lysolipids, Sterolsand Oxidized Lipids, it can't capture the complete spectrum because lipids have a diverse array of chemical structures and a high degree of isomeric overlap.

Our lipid panels cover Ceramide (CER), Cholesteryl Esters (CE), Diacylglycerols (DAG), Dihydroceramide (DCER), Hexosylceramide (HCER), Lactosylceramide (LCER), Lysophosphatidylcholine (LPC), Lysophosphatidylethanolamine (LPE), Monoacylglycerol (MAG), Phosphatidylcholine (PC), Phosphatidylethanolamine (PE), Phosphatidylinositol (PI), Sphingomyelin (SM), Triacylglycerols (TAG). It provides a more in-depth analysis.

We have now added the information in the method section.